# Development and Validation of the PaP Score Nomogram for Terminally Ill Cancer Patients

**DOI:** 10.3390/cancers14102510

**Published:** 2022-05-19

**Authors:** Emanuela Scarpi, Oriana Nanni, Marco Maltoni

**Affiliations:** 1Unit of Biostatistics and Clinical Trials, IRCCS Istituto Romagnolo per lo Studio dei Tumori (IRST) “Dino Amadori”, 47014 Meldola, Italy; oriana.nanni@irst.emr.it; 2Medical Oncology Unit, Department of Specialized, Experimental and Diagnostic Medicine (DIMES), University of Bologna, 40126 Bologna, Italy; marcocesare.maltoni@unibo.it

**Keywords:** cancer patients, end-of-life, hospice, palliative care, prognosis, prognostic score, survival, nomogram

## Abstract

**Simple Summary:**

The Accurate prediction of survival in a palliative care setting is vitally important for clinical, organizational and ethical reasons. Several prognostic tools have been developed to provide reliable estimates of survival in terminally ill cancer patients. We previously developed and validated the Palliative Prognostic (PaP) score, a prognostic tool that assigns patients to three different risk groups according to a 30-day survival probability: group A, >70%; group B, 30–70%; and group C, <30%. As the PaP score does not provide the individual prediction of the survival probability but only probability windows, the present work focuses on the development and validation of the PaP score as a nomogram in order to provide individualized prediction of survival at 15, 30 and 60 days. The nomogram improved the performance of the original PaP score while also maintaining its key feature of simplicity.

**Abstract:**

The validated Palliative Prognostic (PaP) score predicts survival in terminally ill cancer patients, assigning patients to three different risk groups according to a 30-day survival probability: group A, >70%; group B, 30–70%; and group C, <30%. We aimed to develop and validate a PaP nomogram to provide individualized prediction of survival at 15, 30 and 60 days. Three cohorts of consecutive terminally ill cancer patients were used: one (*n* = 519) for nomogram development and internal validation, and a second (*n* = 451) and third (*n* = 549) for external validation. Multivariate analyses included dyspnea, anorexia, Karnofsky performance status, clinical prediction of survival, total white blood count and lymphocyte percentage. The predictive accuracy of the nomogram was determined by Harrell’s concordance index (95% CI), and calibration plots were generated. The nomogram had a concordance index of 0.74 (0.72–0.75) and showed good calibration. The internal validation showed no departures from ideal prediction. The accuracy of the nomogram at 15, 30 and 60 days was 74% (70–77), 89% (85–92) and 72% (68–76) in the external validation cohorts, respectively. The PaP nomogram predicts the individualized estimate of survival and could greatly facilitate clinical care decision-making at the end of life.

## 1. Introduction

The accurate prediction of survival is part of the overall assessment of cancer patients. Information on prognostic factors is useful for a number of reasons. Prognosis helps when making therapeutic decisions, for example, whether or not to continue chemotherapy, when it is time to refer a patient to a palliative care setting, or which palliative setting seems most appropriate. Prognostic awareness facilitates the patient and the caregivers in care planning and is also useful in performing clinical research. Finally, it has an impact on the quality of “care”, that is, on the appropriate utilization of health care resources [1,2].

Although cancer can follow a variety of paths in the early course of the disease, involving a gradual decline over a period of months or years, prognostication in far advanced cancer is easier because there is usually an accelerated decline in the final weeks of life. The challenge facing clinicians caring for advanced patients is to identify the starting point of the irreversible decline.

Despite the predictable pattern of decline, clinical prediction of survival (CPS) is affected by training, experience, seniority and level of acquaintance with the patient and thus risks being inaccurate and over-optimistic [3,4,5,6,7,8,9]. Glare et al. reported that only 61% of cases presented an expected survival accurate to within 4 weeks of actual survival [7]. CPS is easy to use at the bedside, and its accuracy improves with longitudinal repeated measures. The closer to death CPS is, the more accurate it is. There are no sure differences between different specialties and professions. Several studies have reported that clinical experience is important, but this has not been universally confirmed. CPS is less accurate in young people and during the course of a long doctor/patient relationship [8]. Finally, there is debate about the best method for formulating CPS: the temporal way (how long will this patient live?); or the probabilistic way (what is the probability of survival of this patient in a specific time frame?); or a surprise question (would I be surprised if this patient died in a specific time frame?). In the systematic review by White et al. [10], when clinicians were asked to provide a prognostic estimate, accuracy varied from 23% to 78% and differences between predicted and actual survival ranged from −86 days to +93 days. The authors felt that there was sufficient evidence to say that probabilistic estimates of survival may be slightly more accurate than temporal estimates. Overall, evidence suggested that clinicians’ predictions were frequently inaccurate. An inaccurate, overly optimistic prediction has a number of negative consequences. Without improving survival, the presence of a more aggressive diagnostic–therapeutic attitude leads to an increase in futile approaches, late referrals to palliative care settings, and incorrect use of acute healthcare services.

CPS must be integrated by signs and symptoms/objective prognostic factors (i.e., anorexia, weight loss, trouble swallowing and dry mouth, Body Mass Index, weight-loss, phase angle and systemic inflammatory index, etc.) [11,12,13,14]. Prognostic scores or prognostic tools have been identified to increase the prognostic ability. A recent systematic review identified nine prognostic tools that were different in terms of number of factors (too many factors make their use quite complex), such as needing or not needing a blood sample, or integration or no integration between subjective evaluation and objective factors [15]. Some of the scores have been studied and validated by independent groups other than those that built them [16,17,18,19].

We previously developed a Palliative Prognostic (PaP) score for the survival prediction of terminally ill cancer patients [16,20,21,22]. The PaP score consists of a “weighted” scoring system obtained from multivariable analysis that takes into account 6 independent prognostic factors (Karnofsky performance status, clinical prediction of survival, anorexia, dyspnea, total white blood count (WBC) and lymphocyte percentage) selected from 34 clinical and laboratory parameters. This score was repeatedly validated in independent prospective cohorts of patients [22,23,24,25,26,27]. Total scores are in the range of 0–17.5, and patients were assigned to three different risk groups according to a 30-day survival probability: group A (total score 0–5.5), >70%; group B (total score 5.6–11.0), 30–70%; and group C (total score 11.1–17.5), <30. Higher scores predict shorter survival.

With the aim of converting the prognostic categories into meaningful clinical information, we decided to develop a PaP score nomogram. Nomograms are graphical depictions of predictive statistical models for individual patients. They have been developed for various diseases and have shown consistently better performance characteristics than other options [28]. Nomograms also provide a user-friendly interface that does not require computer software for interpretation/prediction [29].

The aim of the present paper was to develop and validate PaP nomogram to provide an individualized prediction of survival of terminally ill cancer patients at 15, 30 and 60 days.

We hypothesized that the application of routinely available survival predictors for terminally ill cancer patients in a nomogram setting could yield prognostic estimates with higher degrees of precision that the original PaP score.

## 2. Materials and Methods

### 2.1. Participants

The original study on the PaP score was conducted on a multicenter Italian cohort of 519 prospectively and consecutively recruited terminally ill cancer patients [16], and this population was used for the development and internal validation of the nomogram. All were adult patients with advanced phase solid tumors, for whom antiblastic therapy was no longer indicated. Palliative hormonal treatment and palliative radiotherapy patients were admitted, whereas patients with renal neoplasms, multiple myeloma and lymphatic pathologies were excluded from the study. Two multicenter Italian validation cohorts were used with the same selection criteria [22,27].

### 2.2. Procedure and Instruments

The PaP score was obtained from a Weibull multivariate regression model, including 6 independent variables (Karnofsky performance status score, CPS, anorexia, dyspnea, total WBC count and lymphocyte percentage), chosen after a backward selection procedure from a set of 34 biological and clinical factors. Each variable is allotted a “partial score” related to the size of the regression coefficient. The sum of the partial scores produced the PaP total score, ranging from 0 to 17.5, and patients were assigned to three different risk groups (A, B and C) according to different 30-day survival probability (Table 1). Higher scores predict shorter survival.

The patients had blood drawn within one week of the PaP score evaluation.

The PaP score was completed on the first day of admission to hospice by the physician.

### 2.3. Statistical Analysis

A nomogram was established on the basis of the results of multivariate analysis using SAS Statistical software version 9.4 (SAS Institute, Cary, NC, USA) and R version 4.0.5 (R Foundation for Statistical Computing, Vienna, Austria; http://www.r-project.org/, accessed on 2 April 2022).

The usefulness of a nomogram is that it maps the predicted probabilities onto points on a scale from 0 to 10 in a user-friendly graphical interface. The total number of points accumulated by the various covariates corresponds to the predicted probability for a patient. The discriminating ability of the PaP score nomogram was assessed using Harrell’s *C*-index for censored data [30], and the bootstrap method was used to account for possible over-fitting. *C*-index is a discrimination measure corresponding to the non-parametric estimate of the area under the receiver operating characteristic curve and can vary from 0.5 (no discrimination) to 1.0 (perfect discrimination).

Calibration plots were generated to explore the performance characteristics of the nomogram at 15, 30 and 60 days, comparing nomogram-predicted survival probability with observed Kaplan–Meier estimates.

The PaP score nomogram was externally validated using two independent cohorts of 451 and 549 terminally ill cancer patients, consecutively entered into hospice programs with the same characteristics (validation cohorts) used to validate the original PaP score [22,27].

The total number of points for each patient in the validation cohort was calculated according to the formulated nomogram. The endpoint was the overall survival (OS), defined as the time from the date of enrollment in the study to the date of death from any cause or the date of the last available information. Survival curves were estimated by the product-limit method of Kaplan–Meier.

The survival database of development and validation cohorts (survival time, status and total PaP score) is available (Appendix A).

## 3. Results

The detailed characteristics of patients in the development and validation cohorts were previously reported [16,22,27], and the three cohorts were similar with respect to all variables. Median age was 67 (range 30–92), 70 (range 21–95) and 71 years (range 18–94) for the development cohort and validation cohorts, respectively. Mean age was 66.2 (standard deviation 12.1), 68.6 (standard deviation 12.5) and 68.8 years (standard deviation 13.2) in the same three cohorts.

Both genders were equally represented, and the most frequent primary cancers were gastrointestinal (37%, 38% and 37%), followed by respiratory (19%, 19% and 19%) and genitourinary (15%, 14% and 18%), in the three cohorts, respectively. Of the patients, 69% had locally advanced disease, and the prevalent metastatic sites were viscera (47%), bone (22%), soft tissue (14%) and central nervous system (11%) in the development cohort and 65%, 56%, 27%, 16% and 13% in the validation cohorts, respectively. Palliative hormonal treatment was administered in 71% of the patients.

The PaP scores in the three cohorts are presented in Table 2. The PaP distributions were quantitatively superimposable (34.3%, 28.2% and 33% in group A; 39.5%, 45.7% and 40.4% in group B; and 26.2%, 26.1% and 26.6% in group C) in the development and validation cohorts, respectively.

The median survival was 32 days (95% CI 29–34) for the development cohort. According to the PaP score, median survival times were 64 days in group A (30-day survival probability, 82.0%), 32 days in group B (30-day survival probability, 52.7%) and 11 days in group C (30-day survival probability, 9.6%).

The prognostic nomogram, integrating all the independent prognostic factors of the PaP score derived from the development cohort, is shown in Figure 1.

The nomogram had a good predictive performance, with a bootstrapped corrected concordance index of 0.74 (95% CI 0.72–0.75). The calibration plot for the probability of survival showed good consistency between the prediction by the nomogram and actual observation (Figure 2A–C).

Median survival in the validation cohorts was 33 (95% CI 29–37, 451 patients) and 22 days (95% CI 19–24, 549 patients). Median survival times were 76 days in group A (95% CI 67–87), 32 days in group B (95% CI 28–39) and 14 days in group C (95% CI 11–18) for the validation cohort of 451 patients. In the validation cohort of 549 patients, the median OS was 59 days (95% CI 52–72) in group A, 18 days (95% CI 16–22) in group B and 6 days (95% CI 5–7) for group C.

The accuracy of the nomogram at 15, 30 and 60 days was 74% (95% CI 70–77), 89% (95% CI 85–92) and 72% (95% CI 68–76), respectively, and the calibration curves fit well between prediction and observation in the probability of OS (Figure 2D–I).

## 4. Discussion

Several prognostic models have been developed to predict survival in terminally ill cancer patients [15,16,17,18,19,31,32,33,34]. However, only a few have been fully evaluated in terms of accuracy (discrimination and calibration) and generalizability (reproducibility and transportability) [15,31]. This may be due to the fact that investigators tend to develop new models rather than comparing and improving existing models [18,19,32,33,34].

In the study by Stone et al. [33], four prognostic tools (Palliative Prognostic Score, Palliative Prognostic Index, Palliative Performance Scale and Feliu Prognostic Nomogram) had acceptable discrimination and calibration. Even if none showed superiority to CPS alone, the authors suggest that tools that perform on par with CPS show enough advantages for CPS, so they recommend their use in clinical practice. They are more objective, more reproducible, useful for second opinions, useful as training and educational tools, helpful in communication, useful for defining entry criteria in clinical studies, and helpful in clinical decision-making. Furthermore, the complete form of the PiPS-B tool [34] showed similar accuracy at both 14 and 56 days to an agreed multi-professional estimate of survival and, therefore, is recommended based on the same reasons of the other scores.

In the study by Hui et al. [35], Subjective Clinical Prediction of Survival (PaP-CPS) worked worse than the objective part of the score (PaP-without CPS). When added to the objective part to make the total score (named PaP total score), it seemed to be less accurate than the objective part alone. The authors suggested that the subjective part of the score (PaP-CPS) added uncertainty to the score itself. On the contrary, a recent study by Yoon et al. [36] compared the accuracy of three prediction models (PaP-CPS, PaP without CPS and PaP total score). PaP total score and PaP-CPS had significantly higher C-indices than the PaP without CPS. No detrimental effect of CPS was seen on objective factors. The reasons for optimal functioning of the PaP Score were: physician characteristics with at least 15 years’ experience; direct use of the original categorical PaP-CPS; study subjects as original PaP study (32 days of median survival in the original study, 33 days in this study). However, due to the PaP Score, we have taken into consideration the need for a better tool, with a continuous presentation of clinical prediction of survival.

In the present study, we evaluated the accuracy and generalizability of the PaP score as a nomogram.

Based on the results from the present study, the PaP score nomogram, which is based on six prognostic factors (Karnofsky performance status, CPS, anorexia, dyspnea, total WBC and lymphocyte percentage), discriminates slightly better than the original PaP score. Calibration plots also demonstrated that the nomograms were well calibrated at 15, 30 and 60 days.

With regard to generalizability, a model’s reproducibility is normally evaluated using bootstrapping techniques in the development sample. The assessment of transportability also requires external validation in different populations. The present study was based on three multicenter databases collected within a wide timeframe, therefore guaranteeing the heterogeneity of the study cohort and making it suitable for transportability assessment. The heterogeneity among centers can thus be considered a strength of the study rather than a weakness. In fact, a high grade of heterogeneity accounting for the entire spectrum of disease, and thus mirroring the clinical world, is needed to generalize a prognostic model.

This study has a number of limitations, the main one being its retrospective design. However, the excellent calibration obtained suggests that this is probably a relatively minor concern. Although the multi-institutional nature of the cohorts could be interpreted as a further limitation, it should be remembered that the heterogeneity of the baseline characteristics, rather than homogeneity, is advisable in order to evaluate the accuracy and generalizability of prognostic models. Another limitation is that the PaP score includes the CPS, a subjective factor varying widely between different observers that often presents a bias towards overestimation [33]. A further criticism is that the score requires a blood sample to be taken [33]. Although laboratory tests can be performed as part of routine clinical practice, they are not very feasible when death is near or patients are hesitant. As reported in previous papers on the PaP Score, blood samples were drawn only as part of routine clinical practice [16,22,37] without providing any method of evaluation or predetermined definition of the symptom in question [37,38], but subsequent articles reported these issues [23].

Moreover, some authors have argued that the PaP score does not provide the individual prediction of the probability of survival but only probability windows [33]. Thus, the present work involved the development and validation of the Palliative Prognostic (PaP) score as a nomogram in order to provide an individualized prediction of survival at 15, 30 and 60 days.

Finally, the PaP score has been criticized because it does not include the “delirium” symptom, which has proven prognostic in other studies [39,40]. However, initial selections of prognostic factors of any score may miss a critical one, making this a universal methodological shortcoming [38]. A study evaluating the impact of this missing symptom in the PaP score showed that the overall performance of the revised PaP score, including delirium, was superimposable with that of the original score, therefore suggesting that no modification was necessary [41].

## 5. Conclusions

Although survival is not the goal of palliative care, physicians and patients need more accurate and generalizable prediction tools to help them in their clinical decision-making.

The PaP score is a tool that assigns patients to three different risk groups of survival probability. In order to determine the probability of survival in this patient setting, even more precisely and accurately, this study was proposed with the aim of converting the three prognostic categories into more accurate clinical information using a personalized determination of the probability of survival by creating a nomogram. The PaP score nomogram had a good predictive performance and calibration and could greatly facilitate clinical care decision-making at the end of life.

In conclusion, the present study helps to better define the general applicability of the PaP score, thus providing an individualized prediction of survival of terminally ill cancer patients at 15, 30 and 60 days.

## Figures and Tables

**Figure 1 cancers-14-02510-f001:**
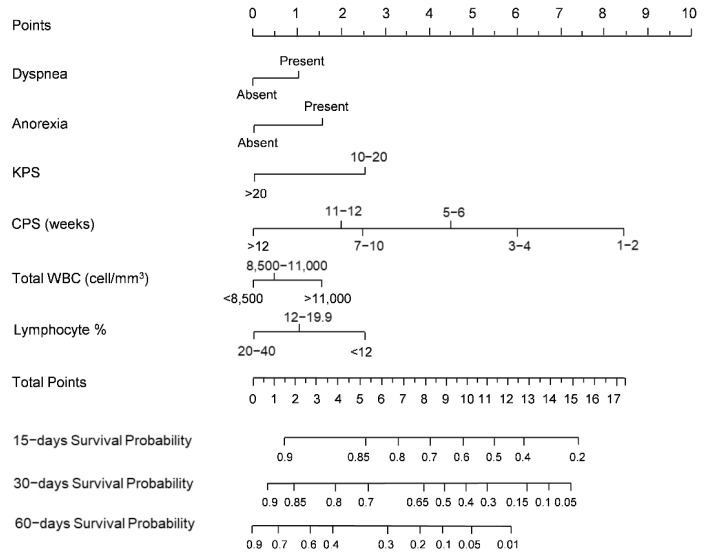
PaP score nomogram (development cohort) predicting 15-, 30-, and 60-day survival. To estimate risk, points were assigned for each independent variable by drawing a line upward form the variable value to the axis labeled “Points”. All points were then summed and plotted on the “Total points” axis; then, a straight line was drawn from the total points axis to the 15-, 30- and 60- days survival axis. KPS: Karnofsky Performance Status; CPS: Clinical Prediction of Survival; Total WBC: Total White Blood Count.

**Figure 2 cancers-14-02510-f002:**
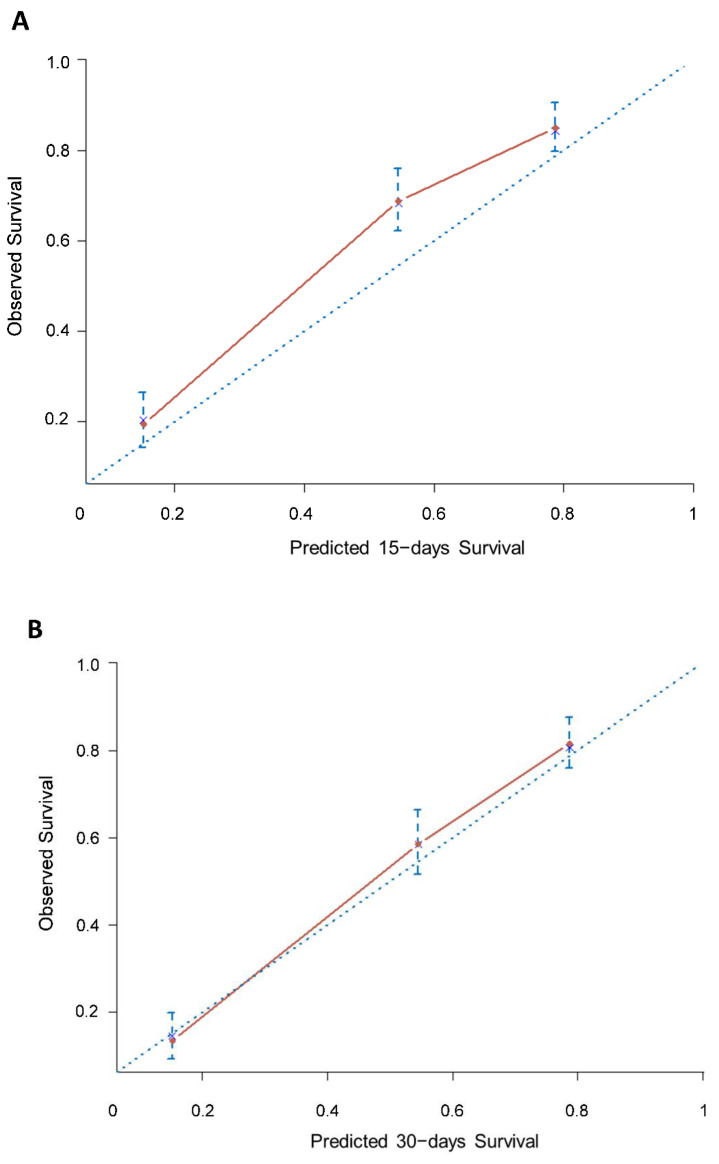
Calibration plots of the PaP score nomogram for 15-, 30- and 60-day survival prediction of the development cohort (**A**–**C**) and validation cohorts of 451 patients (**D**–**F**) and 549 patients (**G**–**I**). The X-axis represents the nomogram-predicted probability of survival; Y-axis represents the observed survival probability (Kaplan–Meier estimates). A perfectly accurate nomogram prediction model would result in a plot where the observed and predicted probabilities for given groups fall along the line. Dots with bars represent nomogram-predicted probabilities and 95% confidence intervals.

**Table 1 cancers-14-02510-t001:** Original PaP score and classification of patients in the three risk groups.

Characteristic	PaP Partial Score
Dypnea	
No	0
Yes	1.0
Anorexia	
No	0
Yes	1.5
Karnofsky performance status	
≥50	0
30–40	0
10–20	2.5
Clinical prediction of survival (weeks)	
>12	0
11–12	2.0
9–10	2.5
7–8	2.5
5–6	4.5
3–4	6.0
1–2	8.5
Total white blood count (cell/mm^3^)	
Normal (4800–8500)	0
High (8501–11000)	0.5
Very high (>11000)	1.5
Lymphocyte rate (%)	
Normal (20.0–40.0)	0
Low (12.0–19.9)	1.0
Very low (0–11.9)	2.5
**Risk groups**	**PaP** **Total score**
A (30-day survival probability >70%)	0.0–5.5
B (30-day survival probability 30–70%)	5.6–11.0
C (30-day survival probability <30%)	11.1–17.5

Total scores range between 0 and 17.5, and patients were assigned to one of three different risk groups according to a 30-day survival probability: group A, <70%; group B, 30–70%; and group C, >30%.

**Table 2 cancers-14-02510-t002:** PaP score in palliative care populations.

	Development Cohort (*n* = 519)	Validation Cohort (*n* = 451)	Validation Cohort (*n* = 549)
Variables	No. Patients (%)	No. Patients (%)	No. Patients (%)
Dyspnea			
No	340 (65.5)	302 (67.0)	367 (66.9)
Yes	179 (34.5)	149 (33.0)	182 (33.1)
Anorexia			
No	191 (36.8)	181 (40.1)	207 (37.7)
Yes	328 (63.2)	270 (59.9)	342 (62.3)
KPS			
≥50	248 (17.8)	140 (31.0)	79 (14.4)
30–40	217 (41.8)	260 (57.6)	356 (64.8)
10–20	54 (10.4)	51 (11.3)	114 (20.8)
CPS (weeks)			
>12	69 (13.3)	49 (10.9)	46 (8.4)
11–12	51 (9.8)	47 (10.4)	52 (9.5)
9–10	41 (7.9)	32 (7.1)	40 (7.3)
7–8	77 (14.8)	64 (14.2)	86 (15.6)
5–6	74 (14.3)	65 (14.4)	78 (14.2)
3–4	109 (21.0)	114 (25.3)	134 (24.4)
1–2	81 (15.6)	80 (17.7)	113 (20.6)
Total WBC (cells/mm^3^)			
Normal (4800–8500)	256 (49.3)	253 (56.1)	252 (45.9)
High (8501–11000)	120 (23.1)	107 (23.7)	101 (18.4)
Very high (>11000)	143 (27.6)	91 (20.2)	196 (35.7)
Lymphocyte rate (%)			
Normal (20.0–40.0)	150 (28.9)	162 (35.9)	114 (20.8)
Low (12.0–19.9)	198 (38.2)	180 (39.9)	148 (27.0)
Very low (0–11.9)	171 (32.9)	109 (24.2)	287 (52.2)
Risk groups			
A (total score 0.0–5.5)	178 (34.3)	127 (28.2)	181 (33.0)
B (total score 5.6–11.0)	205 (39.5)	206 (45.7)	222 (40.4)
C (total score 11.1–17.5)	136 (26.2)	118 (26.1)	146 (26.6)

KPS: Karnofsky Performance Status; CPS: Clinical Prediction of Survival; Total WBC: Total White Blood Count.

## Data Availability

The datasets used and/or analyzed during the current study are available from the corresponding author on reasonable request.

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
