# Peer review of "Development and Validation of the PaP Score Nomogram for Terminally Ill Cancer Patients"

_cancers, 2022, doi:10.3390/cancers14102510_

Round 1

Reviewer 1 Report

Development and validation of the PaP score nomogram for terminally-ill cancer patients

The purpose of the study was to establish a prediction nomograph based on the independent variables of dyspnea, anorexia, Karnofsky performance status, clinical prediction of survival, total white blood count, and lymphocyte percentage in order to classify/stratify terminal cancer patients based on 30 day survival.  The nomogram was tested on two independent cohorts to validate predictability.

Line 18: consider rephrasing “deals with” [see comment for Line 212]

Line 116: The authors are quite familiar with the characteristics of the patient population for this study and I acknowledge that reproducing the entire demographics may be prohibitive but as a stand alone paper, if the authors could simple state the age range, and cancer diagnosis of the three cohorts that would be beneficial for the physician who may not have the time to look up references 6, 12, and 17 to see if this nomogram is applicable to their patient population.  As I am not intimately familiar with the groups previous work and seeing this, it would have been helpful to just have a small section on the characteristics of the population.

Line 208 is confusing to me.  Consider re-wording?

Line 212 “deals with” is slang consider a different word choice. Thus the present work involved developing and validating the Palliative Prognostic (PaP) score as a nomogram in an attempt to provide individualized prediction of survival at 15, 30, and 60 days.

Author Response

Reviewer # 1

Development and validation of the PaP score nomogram for terminally-ill cancer patients

The purpose of the study was to establish a prediction nomograph based on the independent variables of dyspnea, anorexia, Karnofsky performance status, clinical prediction of survival, total white blood count, and lymphocyte percentage in order to classify/stratify terminal cancer patients based on 30 day survival. The nomogram was tested on two independent cohorts to validate predictability.

  • Line 18: consider rephrasing “deals with” [see comment for Line 212]

RE: We have changed the sentence, as suggested by the reviewer.

  • Line 116: The authors are quite familiar with the characteristics of the patient population for this study and I acknowledge that reproducing the entire demographics may be prohibitive but as a stand alone paper, if the authors could simple state the age range, and cancer diagnosis of the three cohorts that would be beneficial for the physician who may not have the time to look up references 6, 12, and 17 to see if this nomogram is applicable to their patient population. As I am not intimately familiar with the groups previous work and seeing this, it would have been helpful to just have a small section on the characteristics of the population.

RE: We thank the reviewer for this suggestion, we have added a brief description of the patient characteristics.

  • Line 208 is confusing to me. Consider re-wording?

RE: According to the reviewer suggestion, we have modified and re-wordering the sentences.

  • Line 212 “deals with” is slang consider a different word choice. Thus the present work involved developing and validating the Palliative Prognostic (PaP) score as a nomogram in an attempt to provide individualized prediction of survival at 15, 30, and 60 days.

RE: We have changed the sentence, as suggested by the reviewer.

Reviewer 2 Report

Excellent paper but line 167-176 is not clear for me; I cannot understand how you created these figures. Do we need the graphs to understand the figures ? I was searching in the graphs how to understand the figures, but I did not find it. 

Are  the group A-B-C (in lines 169-172) the same as the risk groups A-B-C in table 2?

-If not , then  it is very confusing  to use twice group A-B-C in the same paper;

-If yes , it's not possible for me to understand this paragraph

In the graphs are the curves D-I nearly superimposed between calculated and observed survival but the accuracy is only 74%, 89% and 72% for respectively 15-30-60 days: how do you explain that or do I interpret the data wrong??

Author Response

Reviewer # 2

  • Excellent paper but line 167-176 is not clear for me; I cannot understand how you created these figures. Do we need the graphs to understand the figures? I was searching in the graphs how to understand the figures, but I did not find it.

RE: The information on survival according to three risk groups of the PaP score were only reported without figures as this is not the goal of the present paper and these graphs were already published (Pirovano M, JPSM 1999; Maltoni M, JPSM 1999; Maltoni M, Oncologist 2012). For reviewer only, we reported these graphs:

  • Development cohort (n=519)

  • Validation cohort (n=451)

  • Validation cohort (n=549)
  • Are the group A-B-C (in lines 169-172) the same as the risk groups A-B-C in table 2?

-If not, then it is very confusing to use twice group A-B-C in the same paper;

-If yes, it's not possible for me to understand this paragraph

RE: The risk groups A, B and C are the same as the risk groups in table 2. The data reported shows that development and validation cohorts were very similar to each other in terms of median survival and classification of patients in the three risk groups of the PaP score.

  • In the graphs are the curves D-I nearly superimposed between calculated and observed survival but the accuracy is only 74%, 89% and 72% for respectively 15-30-60 days: how do you explain that or do I interpret the data wrong??

RE: The curves in Figure 2 D-I reflect the calibration of the prognostic PaP score, i.e. the extent to which individualized risk estimates provided from the PaP score are reliable. Calibration refers to the accuracy of absolute risk estimates. To the extent that the estimates are accurate, the model is well calibrated.

Discrimination measures how well a prognostic model distinguishes (discriminates) individuals with and without the outcome of interest (dead/alive, event/non-event). It is commonly assessed by Harrell’s C-index.

Reviewer 3 Report

Thank you very much for the work done by the authors of the study. It is very interesting and it can be seen that they have invested a lot of effort in its realization. Below are a number of considerations:

  • The introduction is interesting but brief. The "State-of-the-Art" should be expanded more.
  • The objective stated at the beginning of the method should go in the introduction, just before the hypothesis.
  • The methodological aspects should be more structured. Instead of following a continuous wording, it should be structured in grants.
  • Typically, these sections should be: Procedure, participants, instruments and data analysis.
  • Regarding the results, they are interesting although the format of citation of the Figures should be revised. The name of the Figure usually goes under the image itself unless the Journal stipulates otherwise.
  • It would be interesting to explain in the discussion whether the initial hypothesis is confirmed. It is described very well but it would be interesting if the word hypothesis appeared explicitly.
  • It is recommended to make greater use of the theoretical foundation of the introduction in the discussion.
  • It is recommended to expand the conclusions and make them more concrete. 
  • The references used are correct but scarce. The article should be provided with a greater theoretical foundation. 

In summary, this is a very interesting article. The aspect that should be reviewed in greater depth is the methodological one. Thank you very much for your attention. 

Author Response

Reviewer # 3

Thank you very much for the work done by the authors of the study. It is very interesting and it can be seen that they have invested a lot of effort in its realization. Below are a number of considerations:

  • The introduction is interesting but brief. The "State-of-the-Art" should be expanded more.

RE: As requested, we have expanded the state-of-the-art in the introduction

  • The objective stated at the beginning of the method should go in the introduction, just before the hypothesis.

RE: According to the reviewer’s suggestion, we have move the objective of the study in the introduction section before the hypothesis.

  • The methodological aspects should be more structured. Instead of following a continuous wording, it should be structured in grants.

RE: As suggested, we have more structured the Materials and Methods section.

  • Typically, these sections should be: Procedure, participants, instruments and data analysis.

RE: As suggested, we have more structured the Materials and Methods section

  • Regarding the results, they are interesting although the format of citation of the Figures should be revised. The name of the Figure usually goes under the image itself unless the Journal stipulates otherwise.

RE: We thank the reviewer for this suggestion, we have move the name of the Figure under the image

  • It would be interesting to explain in the discussion whether the initial hypothesis is confirmed. It is described very well but it would be interesting if the word hypothesis appeared explicitly.

RE: We thank the reviewer for this request and we have modified the discussion accordingly.

  • It is recommended to make greater use of the theoretical foundation of the introduction in the discussion.

RE: As recommended, we have changed the discussion accordingly

  • It is recommended to expand the conclusions and make them more concrete. 

RE: As recommended, we have changed the conclusions

  • The references used are correct but scarce. The article should be provided with a greater theoretical foundation. 

RE: According to this suggestion, we have added some references

In summary, this is a very interesting article. The aspect that should be reviewed in greater depth is the methodological one. Thank you very much for your attention. 

Round 2

Reviewer 3 Report

Thank you very much for your effort and time. It can be seen how the manuscript has been improved. I would like to include some considerations and highlight others that have already been made but which may be useful:

  • The introduction could be expanded a little more, as could the number of bibliographic references.
  • The discussion could use more references previously used during the introduction.
  • The participants should be described in more depth in their subsection with their main sociodemographic variables (e.g., age range, mean age and standard deviation).

Thank you for your attention.

Author Response

Thank you very much for your effort and time. It can be seen how the manuscript has been improved. I would like to include some considerations and highlight others that have already been made but which may be useful:

  • The introduction could be expanded a little more, as could the number of bibliographic references.

RE: According to the request, we have sligtly expanded the introduction and added some references.

  • The discussion could use more references previously used during the introduction.

RE: The discussion was modified including some references previously reported in the introduction

  • The participants should be described in more depth in their subsection with their main sociodemographic variables (e.g., age range, mean age and standard deviation).

RE: As requested, we have described the main sociodemographic variables in the Results section and added some information on Materials and Methods (Participants).

Thank you for your attention.
